# Enhancing the Reliability of Intraoperative Ultrasound in Pediatric Space-Occupying Brain Lesions

**DOI:** 10.3390/diagnostics13050971

**Published:** 2023-03-03

**Authors:** Paolo Frassanito, Vito Stifano, Federico Bianchi, Gianpiero Tamburrini, Luca Massimi

**Affiliations:** 1Pediatric Neurosurgery, Fondazione Policlinico Universitario A. Gemelli IRCCS, 00168 Rome, Italy; 2Institute of Neurosurgery, Università Cattolica del Sacro Cuore, 00168 Rome, Italy

**Keywords:** brain tumor, children, extent of resection, intraoperative, real-time imaging, ultrasound

## Abstract

Introduction: Intraoperative ultrasound (IOUS) may aid the resection of space-occupying brain lesions, though technical limits may hinder its reliability. Methods: IOUS (MyLabTwice^®^, Esaote, Italy) with a microconvex probe was utilized in 45 consecutive cases of children with supratentorial space-occupying lesions aiming to localize the lesion (pre-IOUS) and evaluate the extent of resection (EOR, post-IOUS). Technical limits were carefully assessed, and strategies to enhance the reliability of real-time imaging were accordingly proposed. Results: Pre-IOUS allowed us to localize the lesion accurately in all of the cases (16 low-grade gliomas, 12 high-grade gliomas, eight gangliogliomas, seven dysembryoplastic neuroepithelial tumors, five cavernomas, and five other lesions, namely two focal cortical dysplasias, one meningioma, one subependymal giant cell astrocytoma, and one histiocytosis). In 10 deeply located lesions, IOUS with hyperechoic marker, eventually coupled with neuronavigation, was useful to plan the surgical route. In seven cases, the administration of contrast ensured a better definition of the vascular pattern of the tumor. Post-IOUS allowed the evaluation of EOR reliably in small lesions (<2 cm). In large lesions (>2 cm) assessing EOR is hindered by the collapsed surgical cavity, especially when the ventricular system is opened, and by artifacts that may simulate or hide residual tumors. The main strategies to overcome the former limit are inflation of the surgical cavity through pressure irrigation while insonating, and closure of the ventricular opening with Gelfoam before insonating. The strategies to overcome the latter are avoiding the use of hemostatic agents before IOUS and insonating through normal adjacent brain instead of corticotomy. These technical nuances enhanced the reliability of post-IOUS, with a total concordance to postoperative MRI. Indeed, the surgical plan was changed in about 30% of cases, as IOUS showed a residual tumor that was left behind. Conclusion: IOUS ensures reliable real-time imaging in the surgery of space-occupying brain lesions. Limits may be overcome with technical nuances and proper training.

## 1. Introduction

Intraoperative US (IOUS) can be a prompt, effective, low-cost, and reliable adjunct in the surgical resection of space-occupying lesions of the brain. The technology takes advantage of a piezoelectric transducer that produces sound waves starting from electrical signals. These waves are transmitted to the tissues where they are absorbed, reflected, or scattered, based on the acoustic features of the different components of the tissue. The reflected waves are then detected by the same transducer and converted back to electrical signals, which are consequently processed to create the images.

The standard required equipment is quite essential. The technique involves the utilization of a single machine equipped with a transducer probe(s), processing unit, control panel, and a monitor to visualize the images.

Indeed, the IOUS technology can be easily integrated into an existing surgical workflow and is not significantly affected by the type of medical infrastructure, as opposed to more logistically complex and expensive methods, such as intraoperative MRI.

However, ultrasound has been perceived as difficult to learn and standardize. The obtained images have been often considered of low quality, and their consequent interpretation thus can be challenging. Moreover, the intraoperative setting is significantly characterized by certain peculiarities that make the US findings unique. The rate of artifacts is relatively high and may further limit the potential of the technique. However, IOUS has dramatically evolved over the last decade, with vast improvements in image quality and the increasing availability of well-integrated navigation tools.

IOUS has the potential to help surgeons in different ways. First of all, this tool may aid in the localization of intracranial lesions. After the skull opening, the ultrasound probe can be placed onto the dura mater or even directly in contact with the brain to analyze the structures underneath that are not visible. In addition to visualizing the lesion, the surgeon can select and optimize the best surgical corridor to target the lesion for the specific patient, minimizing, for example, the distance from a deep-seated tumor. Moreover, the implementation of different assessment techniques, such as Doppler ultrasonography, enhances the data delivered to the surgeon.

Furthermore, after lesion resection, IOUS may help to assess the extent of resection (EOR), thus detecting residual pathology and reducing the risk of second-look surgery. The surgeon can assess the surgical cavity after the resection is deemed complete and examine its walls looking for, as an example, nodular findings suspicious of residual pathology. However, the IOUS evaluation of the surgical cavity is often limited by the presence of surgical debris and artifacts. In particular, the collapsed surgical cavity can greatly reduce the accuracy of IOUS. Furthermore, IOUS parameters to define surgical margins as disease-free are not clearly stated in the current literature. Most of the studies on IOUS focus on the standard B-mode [1] and suggest that the linear hyperechogenicity may be considered as a physiological reaction to surgical resection, whereas hyperechoic nodules or thickening may suggest a residual lesion.

Despite these obvious advantages, the use of IOUS is partly limited by the quality of US imaging, which is commonly perceived as poor, although modern machines offer improved-quality images compared to the recent past. Another drawback is represented by the lack of overall orientation with respect to anatomic features.

We reviewed our experience with IOUS in the surgery of pediatric supratentorial space-occupying lesions, aiming to focus on the technical limits of this tool and propose strategies and technical nuances to overcome them.

## 2. Materials and Methods

IOUS (MyLabTwice, Esaote, Italy) with a CA1123 curved array microconvex probe was utilized in 45 consecutive pediatric cases with supratentorial space-occupying lesions. This system allows one to also perform neuronavigated ultrasound thanks to a magnetic coregistration of preoperative MR and IOUS. The probe with a 14 mm ray of curvature offers a trapezoidal field of view, wider at depth and narrow at the surface, but allows one to electronically vary the focus to adjust the resolution of insonation over a range of depths, thus combining the advantage of linear and phased-array probes.

A disposable sterile transducer probe cover was routinely used along with sterile transmission gel placed on the probe footprint inside the cover.

IOUS is performed before opening the dura mater to assess the lesion site and its features (pre-IOUS). Brain parenchyma is insonated on planes, which could simplify the comparison with preoperative MR, thus aiding the surgeon’s interpretation of IOUS images.

In deep-seated lesions, IOUS is also used to guide the surgical trajectory.

After the resection of the lesion with microsurgical technique, IOUS is repeated to assess the extent of resection (EOR) on the same planes explored before the resection (post-IOUS).

Contrast agent (SonoVue, Bracco) was administered routinely in pre-IOUS and repeated after surgical resection only in tumor cases.

A single surgeon (PF) acquired all of the IOUS.

Results and technical limits were carefully assessed along with strategies to enhance the reliability of real-time imaging.

## 3. Results

IOUS was successfully performed in all of the cases (23 males and 22 females, median age 6.4 years, ranging from 3 m to 18 years). The microconvex probe ensured the possibility of performing IOUS in all cases, allowing us to access also minimally invasive craniotomic approaches (Table 1). The average number of acquisitions was four IOUS per procedure, with a mean time of 6 min per acquisition.

Pre-IOUS allowed localizing the lesion accurately in all cases, regardless of the type of pathology (16 low-grade gliomas, 12 high-grade gliomas, eight gangliogliomas, seven dysembryoplastic neuroepithelial tumors, five cavernomas, and five other lesions, namely two focal cortical dysplasias, one meningioma, one subependymal giant cell astrocytoma, and one histiocytosis) (Figure 1). In 10 deeply located lesions, the use of IOUS with hyperechoic marker, eventually coupled with neuronavigation, was useful to plan the surgical route.

In seven cases, the administration of contrast ensured a better definition of the vascular pattern of the tumor, while in two cases, it provided a precise definition of the vascular encasement of the vessels of the Willis circle (Figure 2). In this context, the possibility to compare on the same screen the contrast-enhanced image with the standard B-mode image allowed an easy interpretation of this exam. No complication was registered.

In post-IOUS, assessing EOR may be hindered by artifacts that may simulate or hide residual tumors. The main strategies to overcome the former limit are avoiding the use of hemostatic agents before IOUS and insonating through the normal adjacent brain instead of corticotomy.

An additional drawback was noted in 21 cases of large lesions (>2cm). Indeed, the collapsed surgical cavity, especially when the ventricular system is opened, may further complicate the assessment of EOR. This effect is particularly evident in deep-seated lesions, thus affecting the interpretation of IOUS.

On these grounds, we propose to close the ventricular opening, if present, with Gelfoam before insonating. Additionally, we conceived a technical nuance, that we named dynamic IOUS (d-IOUS), to continuously inflate the surgical cavity through pressure irrigation while insonating.

The probe is kept on the brain’s cortical surface, aiming to occlude the corticotomy. The tip of a 14-gauge plastic Abbocath needle (Hospira, Lake Forest, IL, USA) is placed under the US probe into the corticotomy. The Abbocath needle is connected to a syringe filled with 5 mL body-temperature saline. The surgical cavity is irrigated with gentle pressure, thus inflating the surgical cavity and stretching its margins.

This "dynamic" assessment of the surgical cavity allows the reduction of artifacts, in particular of the cavity walls.

Indeed, if hyperechoic nodular images or thickening along the surgical margins that are detected before inflating the cavity are confirmed during the dynamic IOUS, then microsurgical exploration is performed. If hyperechoic nodular images or thickening change to linear hyperechoic images, these may be considered physiological margins of resection without residual tumor (Figure 3, Appendix A).

Thus, this technique enhanced the accuracy of IOUS after lesion resection. There were no procedure-related complications.

In summary, these strategies and technical nuances significantly enhanced the reliability of IOUS. This led to a change of the surgical plan in 30% of cases, as IOUS showed a residual tumor that was left behind, thus avoiding second-look surgery (Figure 4). Moreover, post-IOUS was able to detect intraoperative complications, such as a focal intraparenchymal hemorrhage secondary to brain collapse after the resection of a large intraventricular tumor (Figure 5).

Overall, a total concordance of post-IOUS to postoperative MRI was noted. Indeed, the EOR assessed by MR was consistent with the initial surgical plan and EOR assessed by IOUS. In particular, gross total resection was obtained in 40 cases (89%) and subtotal resection in the remaining five cases due to residual tumor in eloquent region, as highlighted by intraoperative neurophysiological monitoring.

## 4. Discussion

An intraoperative adjunct may be defined as a tool aiding achievement of the maximal safe resection. Indeed, these tools may answer different surgical needs. Neuronavigation is considered the most valid to localize the lesion and delineate a straightforward surgical trajectory. Additionally, the visualization of the lesion could be enhanced by adjuncts, such as fluorescence [2,3,4,5]. Finally, intraoperative imaging is the best means to assess the extent of resection, thus minimizing the risk of second-look surgery [6,7,8,9,10,11]. Although these tools could be complementary, they cannot all be available in the operating room. On the other hand, IOUS is a single tool ensuring ductile applications that may aid in the localization of the lesion, delineation of the surgical trajectory, and assessment of EOR.

Obviously, it is essential to know the advantages and drawbacks of each surgical adjunct in order to confidently rely on its use. Intraoperative MR (IOMR) is considered the gold standard in intraoperative neuroimaging [12,13,14,15,16], although it presents high costs, low availability, and some technical limits [17,18,19,20]. Undoubtedly, IOMR is time-consuming with interruption of surgical flow, thus limiting the possibility of repeating it multiple times during the same surgical procedure [10,21,22]. Furthermore, a recent review of the literature thoroughly questioned the real usefulness of IOMR in estimating the EOR, reporting similar prediction rates compared with IOUS in pediatric brain tumor surgery [23]. From a cost-effectiveness standpoint, IOMR has been found to be still slightly more favorable than IOUS, in terms of gross total resection rates and postoperative performance status, as stated by Mosteiro et al., despite higher global costs and longer surgical times. However, their findings are still preliminary, and further analyses are warranted, above all given the small difference detected between the two techniques [24].

Additionally, IOUS is a reliable, fast, repeatable, and low-cost technique, thus explaining why its application in neurosurgery has become more common over the last few years [25,26,27]. IOUS allows for a real-time update of the intraoperative findings and reliable evaluation of the EOR after lesion resection [17,28]. The potential of IOUS to correctly estimate the EOR is variably reported in the literature, but deemed mostly as satisfying, with good precision and low rates of false-negative or false-positive cases [28,28,29,30].

Notwithstanding the several advantages of IOUS, there are some drawbacks that could be overcome with some technical nuances and tricks (Table 2). Neurosurgeons are not usually well-trained in using the IOUS as well as in interpreting the corresponding images. Although the scanning planes are often different from the classic axial/sagittal/coronal views, we recommend spending some time performing pre-IOUS, aiming to scan the surgical field on planes that could be comparable to preoperative MR in order to maintain a correct orientation.

However, the real-time interpretation of the IOUS images can still be a challenge. For this reason, we think that this technique should become an integral part of neurosurgical training, so that, after a relatively short learning curve, this tool can be employed in the daily surgical routine as a quick and reliable resource.

The role of IOUS in the location of deep-seated lesions is obvious, but even more significant is the aid in locating superficial but not recognizable lesions that are not distinguishable from normal brain parenchyma (e.g., dysembryoplastic neuroepithelial tumor, focal cortical dysplasia). In this context, we prefer to couple the use of IOUS with a 3D reconstruction of FLAIR MR images. This rapid technique offers a reliable image of the brain surface that could easily aid intraoperative orientation also in minimally invasive access [31].

The use of ultrasound contrast agents (UCAs) could improve the definition of the vascular pattern of the tumor. This could eventually correlate with preoperative perfusion MR and grading of the tumor [32,33], though further studies should investigate it.

Furthermore, IOUS after contrast administration may delineate the relationship of the lesion with normal vessels, providing images that could be more easily interpreted compared to echo-color-Doppler mode.

These dyes are by now widespread in adults but are still “off-label” for non-liver applications in children [34], despite a recent statement by the European Federation of Societies for Ultrasound in Medicine and Biology (EFSUMB) encouraging their use in younger patients as well [35].

Localization and definition of the lesion are only a part of the game, in particular when the lesion has a deep location. In this context, the surgical adjunct should ideally guide the surgical route. Neuronavigation is the most widely used tool for this purpose, despite obvious intrinsic limitations. Indeed, neuronavigation is by definition reliable at the start of the procedure, but its reliability may decrease after opening the dura mater due to CSF loss and subsequent brain shift. Retraction of the brain parenchyma to approach deep-seated lesions may further affect neuronavigation. Contrarily, IOUS may provide reliable real-time imaging while advancing through the brain parenchyma. Coupling IOUS with neuronavigation, as warranted by hybrid systems, aids the interpretation of IOUS images and affords the possibility of correcting brain shift using some constant anatomical structures as a spatial reference (e.g., falx, tentorium, ventricles, choroid plexus). Saß B. et al. recently reported their experience with navigated 3D IOUS in glioblastoma surgery. They described the benefits of the technique, highlighting the reliability not only in estimating the EOR, but also in quantifying the brain shift and brain distortion as well as the registration accuracy. The comparison between IOUS images and postoperative MRI underpins the potential of ultrasound to drive the surgical strategy, as seen in one out of five patients for which IOUS detected residual tumor after the resection was initially considered complete, thus leading the surgeon to extend the procedure [36].

Additionally, an echogenic marker, such as Cottonoid patties or Gelfoam, may provide truly real-time anatomical orientation throughout the surgery and delineate the exact surgical trajectory [37]. As the Cottonoid patty could be difficult to differentiate from other hyperechoic materials or structures, such as blood clots or choroid plexus, pulling and releasing their threads could help to distinguish the patties. As an alternative, we use a small piece of Gelfoam, acting as a hyperechoic marker with acoustic shadowing. The direction of the acoustic shadowing and its spatial relationship with the lesion may further help the surgical orientation.

Finally, IOUS may assess the EOR, though several artifacts can hinder the interpretation of the US images, their theoretical reasons being thoroughly described by Selbekk et al. [38]. To reduce these artifacts, different solutions have been proposed. Steno et al. introduced the use of intracavitary mini-probes that, by reducing the distance and the water column between the probe and the tissue to analyze, may decrease the US acoustic enhancement artifacts [39]. This technique is very useful, above all if coupled with a 3D US reconstruction study. However, the width of the field scanned is narrow due to the features of the mini-probe. For this reason, to obtain a 3D US reconstruction, several navigated scans are required, and the process as a result can be time-consuming and cumbersome. Moreover, the insertion of even a small probe into a deep-seated cavity through a narrow corridor can be difficult, and the maneuverability of the probe can be constrained.

A commonly encountered artifact is produced by the passage of the US waves through materials with very different attenuation coefficients. Unsgård et al. reported their experience with a new coupling fluid with the same US attenuation coefficient as the brain. This fluid, avoiding the enhanced brightness given by the saline filling the surgical cavity, could improve the interpretation of the US images after tumor resection, compared to Ringer’s solution [40]. These solutions deserve further investigation but are not commonly used so far. On the other hand, easy and repeatable gestures may reduce the artifacts, such as insonating through normal brain parenchyma instead of corticotomy and insonating before the application of hemostatic agents.

The use of ultrasound contrast agents (UCAs) could improve the sensitivity for residual tumor volume. However, UCAs do not help in assessing the EOR if the lesions do not show contrast enhancement at the IOUS before resection, and they may produce a false negative in case of surgical devascularization of the tumor before UCA administration [35,41,42].

Moreover, IOUS features that define surgical margins as disease-free are unclear to date. A 5 mm rim of hyperechoic tissue on IOUS has been proposed as a reliable predictor of residue [43]. However, the IOUS evaluation of surgical margins may be significantly affected if the cavity collapses or the margins fold upon themselves and all of the above-mentioned proposed solutions fail to resolve this problem.

In our experience, the dIOUS is a simple and effective technique that allows distending the cavity walls, thus reducing the artifacts and enhancing the accuracy of IOUS to assess the surgical cavity when it collapses. As a consequence, the IOUS sensitivity and specificity in assessing the EOR is increased in our experience and shows a total concordance with postoperative MRI.

## 5. Conclusions

IOUS ensures reliable real-time imaging in the surgery of space-occupying brain lesions. Limits may be overcome with technical nuances and proper training.

Although further studies are needed to better validate these strategies in current neurosurgical practice, more widespread adoption of this tool is advisable.

## Figures and Tables

**Figure 1 diagnostics-13-00971-f001:**
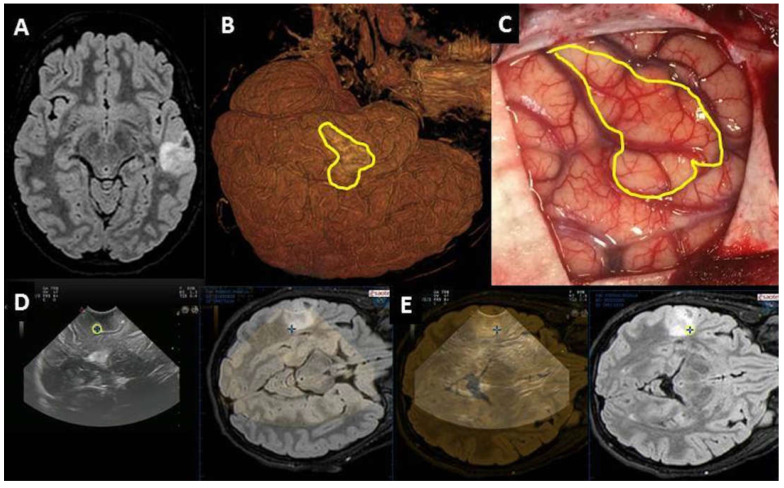
Left temporal DNT documented by MRI (**A**). The 3D reconstruction of brain surface (**B**) is consistent with intraoperative anatomy (**C**); intraoperatively, the tumor was not clearly distinguishable from the normal brain parenchyma. Neuronavigated pre-IOUS showing the hypoechoic cortical tumor (**D**). The location of the tumor was confirmed by superimposing the preoperative MRI onto the IOUS image (**E**).

**Figure 2 diagnostics-13-00971-f002:**
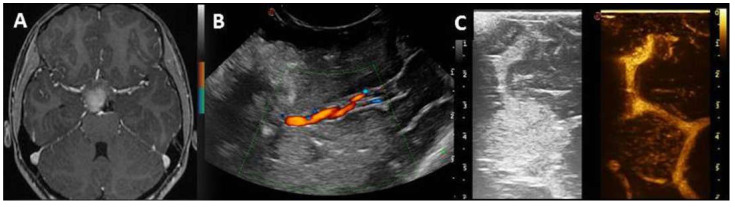
Axial T1-weighted MR after contrast administration showing diencephalic low-grade glioma (**A**). Although the spatial relationship of the tumor with the vessel of Willis circle could be explored through echo-color-Doppler mode (**B**), IOUS post-UCA administration provides a better and more easily interpretable picture (**C**).

**Figure 3 diagnostics-13-00971-f003:**
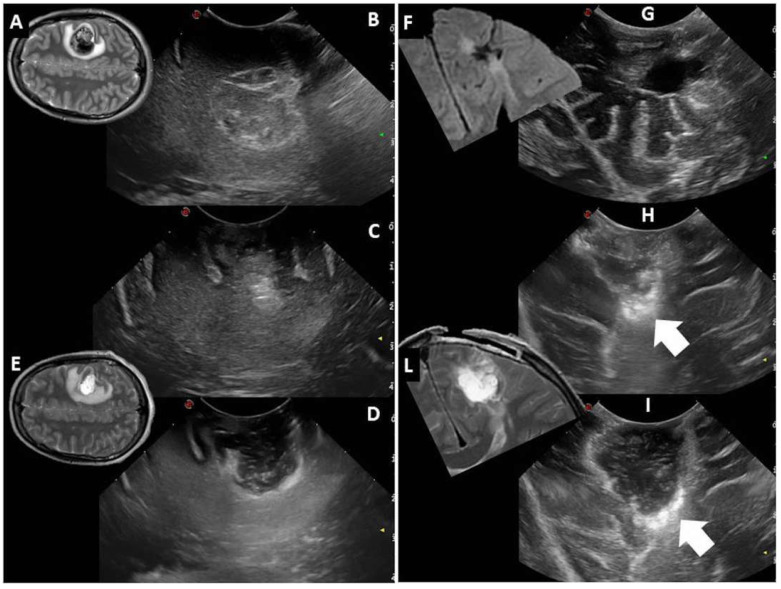
Case #1 (**left**): Right parietal cavernoma documented by preoperative MR (**A**) and IOUS before surgical resection (**B**). After resection of the lesion, IOUS visualization of the collapsed surgical cavity is limited by surgical artifacts (**C**). The dIOUS allows the distension of the surgical cavity, which is bordered by linear hyperechoic artifacts (**D**). The complete resection of the lesion is concordant with postoperative MR (**E**). Case #2 (**right**): Left parietal glioneuronal tumor documented by MR (**F**) and IOUS before surgical resection (**G**). After surgical resection of the tumor, IOUS shows a possible residual tumor (**H**, arrow). The dIOUS rules out the presence of residual tumor, as the hyperechoic thickening becomes linear during gentle pressure irrigation (**I**, arrow). Radical resection is confirmed by postoperative MR (**L**).

**Figure 4 diagnostics-13-00971-f004:**
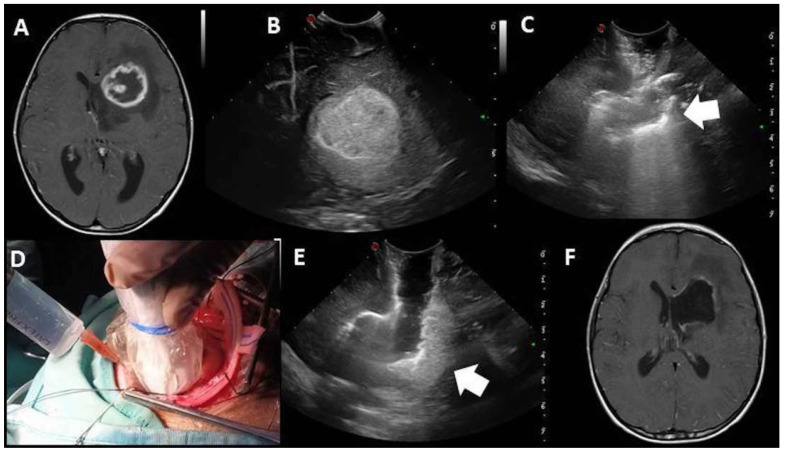
MR showing large frontal periventricular high-grade glioma (**A**). Pre-IOUS confirming large hyperechoic nodular lesion, consistent with preoperative MR picture (**B**). Post-IOUS is limited by artifacts (arrow) and partially collapsed surgical cavity (**C**). The d-IOUS (**D**) ensures distension of the surgical cavity with reduction of the artifacts, thus confirming the presence of residual tumor (arrow, **E**) that was then surgically resected. Radical resection was confirmed by postoperative MR (**F**).

**Figure 5 diagnostics-13-00971-f005:**
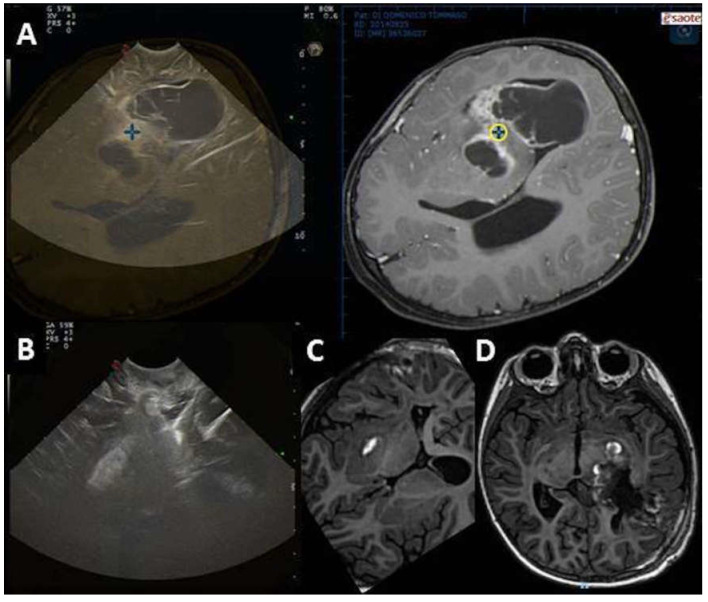
Neuronavigated Pre-IOUS showing large intraventricular tumor (**A**). Post-IOUS documented a focal thalamic hemorrhage (**B**). Postoperative MR confirmed this finding (**C**) and the complete resection of the tumor (**D**).

**Table 1 diagnostics-13-00971-t001:** Demographic and clinical data.

Number of cases	45
M:F	23:22
Age	3m–18 years (median 6.4 years)
Pathology	16 LG gliomas 12 HG gliomas 8 gangliogliomas 7 DNTs 5 cavernomas 5 other (2 FCDs, 1 meningioma, 1 SEGA, 1 histiocytosis)
Size	24 (<2 cm) 21 (>2 cm)
Site	15 temporal > 8 frontal > 6 parietal > 5 intraventricular > 4 occipital–4 diencephalic > 2 thalamic > 1 insular

LG: low grade, HG: high grade, DNT: dysembryoplastic neuroepithelial tumor, FCD: focal cortical dysplasia, SEGA: subependymal giant cell astrocytoma.

**Table 2 diagnostics-13-00971-t002:** Technical limits of IOUS and strategies to enhance its reliability.

Pre-IOUS	Problems	Solutions
Localization	Deep-seated lesion	IOUS coupled with neuronavigation
Localization	Superficial lesion not distinguishable from brain parenchyma	IOUS coupled with MR 3D reconstruction (FLAIR)
Definition	Vascular relationship	IOUS + UCA Echo-color-Doppler mode
Definition	Vascular pattern	IOUS + UCA
Surgical route	Deep-seated lesion	IOUS coupled with neuronavigation (hybrid system) IOUS with intracavitary hyperechoic marker
**Post-IOUS**		
EOR	Collapsed surgical cavity	Inflation of the surgical cavity through pressure irrigation while insonating *(Dynamic IOUS, see text for further details)*
EOR	Open ventricle	Closure of the ventricular opening with Gelfoam before insonating
EOR	Artifacts	Avoiding the use of hemostatic agents before IOUS Insonating through normal adjacent brain instead of corticotomy

IOUS: intraoperative ultrasound, UCA: ultrasound contrast agent, EOR: extent of resection.

## Data Availability

Research data are available on request.

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
