# Peer review of "Enhancing the Reliability of Intraoperative Ultrasound in Pediatric Space-Occupying Brain Lesions"

_diagnostics, 2023, doi:10.3390/diagnostics13050971_

Round 1
Reviewer 1 Report
The authors have prepared an exciting manuscript on the reliability of ioUS in pediatric tumor pathology.
The paper is very well written and has remarkable clinical relevance.
The authors adequately describe the limitations of the technique and propose ways to overcome the limitations during image acquisition.
To improve the quality of the article, I suggest adding details of the ultrasound scan and probe, such as the footprint, frequency, FOV, 2D or 3D, method of sterilization, the average number of acquisitions in each intervention, and clarifying if one single surgeon acquired all the preoperative images. I would also suggest giving details about the pre and postoperative volumetry of the lesions. Do not just limit to dichotomizing between < or > 2cm. Another question to clarify is whether all the cases were gross total resections or not. Finally, an overlayed of ultrasound and MR images is mentioned. Is it a real fusion of both modalities? Is any navigation system coupled to ultrasound? Is it routinely applied?
Author Response
We kindly thank the reviewer for the comments and the opportunity to improve the manuscript.
Technical details of the probe and the neuronavigated US system have been included in the methods section. Actually, it is a neurovigated system allowing magnetic coregistration of preoperative MR with IOUS, with the possibility of fusing the MR and IOUS images.
A single surgeon acquired IOUS.
Volumetry of the lesions is unfortunately unavailable in all the cases. However, we find the dichotomy quite useful for the purpose of the paper, that is providing a sort of practical guide for IOUS.
The rate of gross total resection has been included in the results section.
Reviewer 2 Report
Authors present a retrospective study on 45 consecutive pediatric cases who underwent surgery for supratentorial lesions (mostly gliomas) to asses the application of intraoperative ultrasound (ious) for resection. . In 10 deeply located lesions, IOUS with hyperechoic marker, eventually coupled with neuronavigation, was useful to plan the surgical route; in 7 cases the administration of contrast warranted a better definition of the vascular pattern of the tumor; post-IOUS allowed the evaluation of extent of resection (EOR) reliably in small lesions (< 2cm), whereas in large lesions (>2cm) assessing EOR is hindered by the collapsed surgical cavity and artifacts, especially when the ventricular system is opened.
Low number of patients and retrospective character of the study as well as different pathologies are drawbacks of the study. Introduction is too short and needs to be expanded on use of intraoperative ultrasound for resection of supratentorial lesions and in pediatric neurosurgery.
Materials and Methods are poorly written and do not provide enough information. Did you acquire 3D Dataset using ultrasound, was this Dataset fused with preoperative MRI? Was US navigated? How did you measure spatial overlap in pre-resection and post-resection ultrasound - Euclidian offset? Dice coefficient? Hausdorff distance? Did you use Doppler? Please clarify.
Results: Table - Size >< 2 cm ... 2 cm axial diameter? I suggest to include a detailed Table with all patients, their baseline data, histology, size of the tumor using volumetry (in cm3).
I suggest to expand illustrative cases and add preoperative and postoperative imaging. Furthermore, artifacts which limit the estimation of extent of resection can be minimized by use of navigated ultrasound. It is unclear what is the added value of a study with non-navigated ultrasound, please elaborate. Which problems did you encounter with application of contrast? For navigation - did you used fiducial-based registration or intraoperative imaging. If you claim that there was good overlap between intraoperative post-resection IUOS and postoperative MRI - on which grounds is this claim based? Did you fuse 3D ultrasound with postoperative MRI, or this is just an estimation of the surgeon? How many surgeons were involved as primary operators? What was the complication rate?
For Discussion I suggest to include and extensively comment:
Saß B, Zivkovic D, Pojskic M, Nimsky C, Bopp MHA. Navigated Intraoperative 3D Ultrasound in Glioblastoma Surgery: Analysis of Imaging Features and Impact on Extent of Resection. Front Neurosci. 2022 May 9;16:883584. doi: 10.3389/fnins.2022.883584. PMID: 35615280; PMCID: PMC9124826.
Šteňo A, Buvala J, Šteňo J. Large Residual Pilocytic Astrocytoma After Failed Ultrasound-Guided Resection: Intraoperative Ultrasound Limitations Require Special Attention. World Neurosurg. 2021 Jun;150:140-143. doi: 10.1016/j.wneu.2021.03.138. Epub 2021 Apr 2. PMID: 33819702.
Several comments on cost effectiveness:
Mosteiro A, Di Somma A, Ramos PR, Ferrés A, De Rosa A, González-Ortiz S, Enseñat J, González JJ. Is intraoperative ultrasound more efficient than magnetic resonance in neurosurgical oncology? An exploratory cost-effectiveness analysis. Front Oncol. 2022 Oct 28;12:1016264. doi: 10.3389/fonc.2022.1016264 Conclusions are too short and do not follow the results.Author Response
We kindly thank the reviewer for the comments and the opportunity to revise and improve the paper.
We agree that low number of patients and retrosepctive nature are drawbacks of the study. On the other hand, we think that the wide range of different pathologies further proves the usefullness of IOUS.
Introduction has been extended, as suggested.
Material and methods section has been implemented. We used a hybrid system coupling US with magnetic navigation. The system allows coregistration of the preoperative MR with intraoperative US.
Concerning the results section, volumetry of the tumor is not available. However, we found the dichotomy < and >2cm useful as practical rule for IOUS use. Tables 1 summarizes the features of patients included in the study while table 2 summarizes the "lesson learned" from the different cases. We think that a third table including the characteristcs of all the cases would make the paper more sloppy.
The discussion has been extended and references list accordingly modified.
Round 2
Reviewer 2 Report
Authors have sufficiently responded to reviewers remarks.